# Heart Failure in Type 1 Diabetes: A Complication of Concern? A Narrative Review

**DOI:** 10.3390/jcm10194497

**Published:** 2021-09-29

**Authors:** Ana María Gómez-Perez, Miguel Damas-Fuentes, Isabel Cornejo-Pareja, Francisco J. Tinahones

**Affiliations:** 1Department of Endocrinology and Nutrition, Virgen de la Victoria University Hospital, 29010 Málaga, Spain; anamgp86@gmail.com (A.M.G.-P.); migueldamasf@hotmail.com (M.D.-F.); fjtinahones@hotmail.com (F.J.T.); 2Instituto de Investigación Biomédica de Málaga (IBIMA), Cellular and Molecular Endocrinology, Virgen de la Victoria University Hospital, 29010 Málaga, Spain; 3Spanish Biomedical Research Center in Physiopathology of Obesity and Nutrition (CIBERObn), Instituto de Salud Carlos III, 28029 Madrid, Spain

**Keywords:** type 1 diabetes, heart failure, cardiovascular disease, diabetic myocardiopathy

## Abstract

Heart failure (HF) has been a hot topic in diabetology in the last few years, mainly due to the central role of sodium-glucose cotransporter 2 inhibitors (iSGLT2) in the prevention and treatment of cardiovascular disease and heart failure. It is well known that HF is a common complication in diabetes. However, most of the knowledge about it and the evidence of cardiovascular safety trials with antidiabetic drugs refer to type 2 diabetes (T2D). The epidemiology, etiology, and pathophysiology of HF in type 1 diabetes (T1D) is still not well studied, though there are emerging data about it since life expectancy for T1D has increased in the last decades and there are more elderly patients with T1D. The association of T1D and HF confers a worse prognosis than in T2D, thus it is important to investigate the characteristics, risk factors, and pathophysiology of this disease in order to effectively design prevention strategies and therapeutic tools.

## 1. Introduction

Heart failure (HF) is one of the most frequent causes of hospital admission and has a poor prognosis in most cases, despite great pharmacological advances developed in recent decades for heart failure with reduced ejection fraction (HFrEF; left ventricular ejection fraction < 40%) [1]. In addition, there is a group of patients with an ejection fraction > 50% (heart failure with preserved ejection fraction or HFpEF), who also have a poor prognosis, but for whom there are still no proven effective therapies [2]. In the field of diabetes, thanks to the role of several pharmacological groups in the prevention of cardiovascular events, among which is hospitalization for HF, this condition has acquired a central role as one of the most frequent complications of type 2 diabetes (T2D) [3]. However, it is also gaining more interest in patients with type 1 diabetes (T1D), especially due to the longer life expectancy that makes it easier to find older patients with long-standing T1D. In fact, epidemiological evidence shows us that diabetes increases the risk of HF twice in men and up to five times in women [4]. Among patients with diabetes, an estimated 40% have HF, with higher mortality and risk of hospitalization than patients without diabetes [5].

Although due to the development of sodium-glucose cotransporter 2 inhibitors (iSGLT2) and glucagon-like peptide 1 analogues (aGLP1), the evidence on heart failure in T2D is growing and extensive, in T1D there is a paucity of data. In this review, we will focus on the evidence about HF in T1D with a special interest in the epidemiology, risk factors, and pathophysiology of this complication.

## 2. Epidemiology and Risk Factors

### 2.1. Epidemiology

Until a few years ago, most epidemiological and long-term follow-up studies on diabetes were focused on classic cardiovascular complications with an atherosclerotic profile. However, thanks to the data from the cardiovascular safety studies of new antidiabetic drugs, more and more studies with epidemiological data are being published also on T1D, though its incidence and prevalence are not well established. In a 10-year retrospective study performed by McAllister et al. of over 3.25 million people without DM and with T2D and T1D, there were 1313 events of HF among patients with T1D. The crude incidence rate of hospitalization for HF in the T1D group was 5.6 per 1000 person-years compared to 2.4 in patients without diabetes and 12.4 in patients with T2D. However, the case-fatality rate was higher in patients with T1D than people without diabetes mellitus; the difference was larger for men (OR, 1.91; 95% CI, 1.68–2.18) than for women (OR, 1.31; 95% CI, 1.05–1.65) [6].

In a recent observational study, Kristófi et al. analyzed a population of 59,331 patients with T1D and 484,241 patients with T2D in Sweden and Norway, looking for the prevalence and event rates of myocardial infarction, HF, stroke, chronic kidney disease, all-cause death, and cardiovascular death. They observed that patients with T1D had a higher risk of HF and renal disease in different age groups than patients with T2D. The age-adjusted risk for patients 65–79 years showed that the risk of heart failure was 1.3 to 1.4 times higher in patients with T1D than with T2D. They also found greater cardiovascular mortality in T1D in patients above 55 years [7]. Similarly, in a recent meta-analysis carried out by Cai et al. that included 10 observational studies with 166,027 patients, a relative risk of heart failure of 4.29 (95% CI 3.42–4.86) was observed in patients with T1D compared with healthy controls. This meta-analysis suggests that T1D is associated with an increased risk of several cardiovascular diseases, among them HF [8].

In another recent paper, Chadalavada et al. investigated the effect of diabetes in mortality and incident HF with the entire population of the UK Biobank. They included a total population of 493,167 participants, of which 22,685 had diabetes (4.6%). They found a hazard ratio (HR) for HF of 1.9 (CI 95% 1.7–2) among patients with diabetes compared to healthy controls. Interestingly, they found that women with T1D had an 88% increased risk of HF compared to men (HR 4.7 (CI 95% 3.6-6.2) vs 2.5 (CI 95% 2.0–3.0), respectively) and this association was independent of confounding factors. In T2D, the risk of HF was also greater in women but to a lesser extent [9]. On the incidence of HF in T1D, Avogaro et al. performed a systematic review and meta-analysis, including 6 studies published between 1990 and 2018. In their age-adjusted model, the incidence rate of HF in patients with T1D was 3.18 (*p* < 0.001) compared to the general population [10]. Finally, in a nationwide retrospective study performed in Korea, Lee et al. explored HR for cardiovascular disease and early death in people with T1D compared with people with T2D and healthy controls. During more than 93,300,000 person-years of follow-up, they found an HR of hospitalization for HF of 2.105 (CI 95% 1.901–2.330) in T1D compared to T2D and an HR of 3.024 (CI 95% 2.730–3.350) compared to the non-diabetes group. This greater risk for HF in T1D remained after adjustment for fasting plasma glucose and some cardiovascular risk factors, such as smoking, dyslipidemia, hypertension, physical activity, or body mass index, among others [11].

### 2.2. Risk Factors

Cardiovascular risk factors are well established and data from the Diabetes Control and Complications Trial (DCCT) and its observational follow-up Epidemiology of Diabetes Interventions and Complications (EDIC) demonstrated that an early period of 6–7 years of intensive glycemic control significantly reduced the risk for cardiovascular disease. Although the effect tends to decrease over the years, it remains highly significant 30 years later (a reduction of 30% compared to conventional treatment; *p* = 0.016) [12,13]. However, the presence of diastolic dysfunction has been demonstrated even in adolescent and young adult patients with T1D, as potential early markers of heart failure [14,15]. Moreover, some studies suggest that multiple risk factor control reduces the risk for myocardial infarction or stroke but has little association with the risk for HF in T1D [16]. Therefore, the study and identification of risk factors for HF in people with T1D are key points to improve the detection, management, and prognosis of this complication. 

Regarding glycemic control, the data from DCCT/EDIC studies showed that glycemic control represented as HbA1c was the strongest modifiable risk factor for congestive heart failure after 29 years of follow-up in 1441 patients with T1D. Each 1% increase in HbA1c produced an incidence ratio of 3.15 (*p* < 0.01) for congestive heart failure [12]. Therefore, early intensive therapy seems to have an effect in reducing HF risk in the long-term (five-fold difference among intensive therapy group and conventional treatment), though in the analysis at 30 years, the number of events related to HF were too small to establish a definitive conclusion [13]. In line with these results, Rawshani et al. assessed the relative prognostic importance of 17 risk factors on cardiovascular outcomes in a nationwide register of patients with diabetes in Sweden. For HF, they found that the most important predictors were albuminuria (β-coefficient 3.63 (3.05–4.31)), HbA1c (β-coefficient 1.025 (1.020–1.030)), and systolic blood pressure ((β-coefficient 1.35 (1.25–1.44)) [17]. Kristófi et al. [7], in line with previously published data that identified albuminuria and chronic kidney disease (CKD) as risk factors for cardiovascular disease in T1D [18,19], found that prevalent and incident CKD are more common in T1D than in T2D. These higher levels of kidney impairment may play a role in higher rates of cardiovascular disease, reinforcing the importance of cardiorenal syndrome.

A possible explanation for this greater risk of renal and cardiovascular disease, and among them HF, is that the disease duration is often longer in T1D than T2D. Therefore, the probability of microvascular complications and the effects of hyperglycemia in cardiovascular outcomes are greater in T1D [7]. In fact, in the meta-regression analysis performed by Avogaro et al., age was significantly associated with the incidence ratio of HF [10]. Another study by McAllister and colleagues of over 3.25 million people among which there were 18,240 subjects with T1D, found that by the age of 20 years the prevalence of HF is similar among patients with T1D and T2D, but by the age of 80 years the prevalence of HF is higher in T1D and the same occurs with case-fatality rate, suggesting that the accumulation of risk factors and more prevalent microvascular complications in this group may contribute to higher incident and prevalent HF. Another hypothesis suggested in this study was that lower rates of prescription drugs known to reduce the risk of HF, such as antihypertensives, drugs acting on the renin-angiotensin system, or lipid lowering drugs, may also contribute to higher incident and prevalent HF. However, differences remained despite the older mean age of patients with T2D and after adjusting for individual risk of HF according to baseline characteristics. Though these are data from a retrospective study and some information was taken from clinical recordings with a risk of missing information, it is a very interesting hypothesis to consider [6].

Another emerging field of investigation in HF in T1D that may be intimately linked to glycemic control is cardiac autoimmunity. In an analysis derived from DCCT/EDIC, Sousa et al. measured the prevalence and profiles of cardiac autoantibodies in samples from DCCT and divided them in two groups, patients with HbA1c > 9% (n = 83) and patients with HbA1c < 7% (n = 83) at 26 years of follow-up. The same analysis was performed in similar groups of patients with T2D. They found that the DCCT HbA1c > 9% group had significantly higher levels of cardiac autoantibodies than the DCCT HbA1 < 7% group, while glycemic control was not related to cardiac autoimmunity in T2D. Moreover, positivity for two or more autoantibodies during DCCT was associated with a greater risk of cardiovascular disease (HR 16.1 [95% CI 3.0–88.2]) and coronary artery calcification (OR 60.1 [95% CI 8.4–410.0]) [20]. The same authors recently published a study showing that cardiac autoimmunity is related to subclinical myocardial dysfunction, independent of classical cardiovascular disease risk factors. They observed in a sample from DCCT that patients with two or more cardiac autoantibodies had greater left ventricular end-diastolic volume, end-systolic volume, left ventricular mass, and lower left ventricular ejection fraction [21]. These observations suggest that there are different mechanisms for cardiovascular disease and thus for HF in T1D and T2D and those mechanisms may be tightly related to long exposure to hyperglycemia in T1D.

Finally, regarding risk factors, there is an interesting study carried out by Khedr et al. on 78 adolescents with T1D of at least 6 years of duration, in whom they analyzed some lipid biomarkers as predictors of diastolic dysfunction. They found diastolic failure to occur in 50% of the females and 66.6% of the males, and described that lower high-density lipoproteins (HDL) (OR 0.93, 95% CI 0.88–0.99) and a higher total cholesterol/HDL ratio (OR 2.55, 95% CI 1.9–5.45) and triglycerides/HDL ratio (OR 2.74, 95% CI 1.12–6.71) were associated with diastolic failure [22].

Considering that classical risk factors seem to be important, but it also seems that HF in T1D has differential characteristics, it is necessary to continue investigating the most important risk factors for the development of this complication (Figure 1).

## 3. Pathophysiology

The mechanisms responsible for the association between diabetes and heart failure are not entirely clear, although a great variety of them have been proposed, such as endothelial dysfunction, alterations in glucose and fatty acid metabolism at the myocardial level, myocardial fibrosis, the increase in oxidative stress, or the activation of local neuro-hormonal systems, such as the renin-angiotensin-aldosterone system, endothelin, or the sympathetic nervous system [23] (Figure 1). It has also been proposed that some of these mechanisms can cause systolic or diastolic ventricular dysfunction even in the absence of coronary artery disease or structural disease [5]. There is a multitude of preclinical data, but they are still to be clarified.

Diabetic cardiomyopathy pathophysiology is widely studied in T2D, while its mechanisms in T1D are less clear. Hyperglycemia and chronic inflammation present in both types, promoting cardiac hypertrophy and fibrosis, increasing myocardial stiffness, and resulting in diastolic and systolic dysfunction. Increased levels of glucose lead to a higher production of advanced glycation end products (AGEs), which have been suggested to trigger deleterious effects on ventricular function through the formation of crosslinks between collagen molecules in the extracellular matrix, impairing its degradation and leading to myocardial stiffness and diastolic dysfunction [24]. Activated endothelial cells also contribute by promoting the uncoupling of endothelial nitric oxide synthase (NOS) resulting in diminished nitric oxide (NO) levels. This decreases soluble guanylate cyclase (sGC) activity and cyclic guanosine monophosphate (cGMP) content in the myocardium, which impairs the protective effects of protein kinase G (PKG) [25].

Due to the insulinopenia in T1DM, fatty acid β-oxidation is increased to maintain adenosine triphosphate (ATP) producti; however, this process becomes ineffective during diabetes evolution, resulting in intracellular lipid accumulation and lipotoxicity [26]. Increased intracellular fatty acid concentration and mitochondrial dysfunction lead to an increased production of reactive oxygen species (ROS). Excess ROS production causes the activation of cellular and mitochondrial nitrogen oxides (NOX), which leads to the generation of superoxide and hydrogen peroxide [27]. These effects result in cardiomyocyte loss, cardiac hypertrophy, and inflammation with fibrosis of the extracellular matrix [28]. N-acetylcysteine (NAC) has been used as an antioxidant in mouse models of T1D to normalize oxidative stress and therefore prevent the development of cardiomyopathy [29].

Mitochondrial dysfunction is usually found in cardiac tissue in T1D patients. Decreased mitochondrial oxidative capacity is caused by altered mitochondrial ultrastructure, proteomic remodeling, and oxidative damage to proteins and mitochondrial DNA [30].

Concerning cardiac inflammation, the infiltration of macrophages and lymphocytes is usual in DM. These inflammatory cells secrete cytokines, such as tumor necrosis factor (TNF), interleukin 6 (IL-6), interleukin 1β (IL-1β), interferon-γ, and transforming growth factor β (TGFβ) that can produce profibrotic responses, leading to further adverse remodeling. Studies in mice have detected higher T-cell infiltration in the myocardium in T1D [31] and some attempts to reduce cardiac fibrosis by decreasing T-cell trafficking have been successful [32]. Regarding the immune system, as we mention before, Sousa et al. observed higher levels of cardiac autoantibodies in patients with T1D and poor glycemic control, and patients positive for ≥2 cardiac autoantibodies were more likely to have subclinical myocardial dysfunction as well as a higher cardiovascular disease risk. Chronic hyperglycemia may cause subclinical myocardial injury favoring the exposure of heart muscle proteins as α-myosin to the immune system. In patients with T1D and poor glycemic control, the immune system is dysregulated and may overreact to these proteins, producing an expansion of proinflammatory CD4 T-cells specific to α-myosin and the development of autoantibodies [20,21].

Another mechanism implicated in the pathophysiology in T1D mouse models is increased cardiomyocyte intracellular Ca^2+^ due to lower sarcoplasmic reticulum Ca^2+^ pump activity because of the decreased glucose transporter type 4 (GLUT 4) recruitment to the plasma membrane, mediating this disturbance in contractile dysfunction and arrhythmia [33].

Renin-Angiotensin-Aldosterone activity is increased under diabetic conditions. Angiotensin-II receptor type 1 (AT1R) density and synthesis are increased in T1D hearts, and the increase in fibrosis is partially inhibited following treatment with ACE inhibitors and AT receptor blockers [34]. Moreover, a frequent complication related to sustained hyperglycemia is cardiac autonomic neuropathy which includes abnormalities in heart rate control, vascular hemodynamics, and cardiac structure and function. An early characteristic of cardiac autonomic neuropathy is the reduction of parasympathetic activity with an imbalance toward higher sympathetic activity [35].

Among the new fields that are opening in the pathophysiology of heart failure, the intestinal microbiota and some of its metabolites stand out [36]. In some models, *Akkermansia*, *Prevotella 9*, *Paraprevoltella*, and *Phascolarctobaterium* have been associated with changes in cardiac structure and function [37]. The "intestinal hypothesis" of heart failure postulates that the reduction in cardiac output causes damage to the intestinal barrier that generates dysbiosis, favoring the proliferation of pathogenic species such as *Candida* and the reduction of anti-inflammatory bacteria such as *Faecalibacterium prausnitzii.* Similarly, the microbiota can promote heart failure through the modulation of intestinal immunity. Segmented filamentous bacteria favor the production of IL-6 and interleukin 23 [38] and *Bacteroides Fragilis* favors the production of anti-inflammatory cytokines that, in murine models, have been shown to reduce ventricular remodeling after myocardial infarction [39]. Bacterial metabolites also seem to have a role; for example, the reduction of short-chain fatty acids can favor the damage of the intestinal barrier and promote dysbiosis and the translocation of endotoxins to the bloodstream [40]. Trimethylamine N-oxide (TMAO) also seems to act as a risk factor for heart failure, since it has been observed in animal models to facilitate the release of calcium in the heart muscle by altering contractility and may also increase myocardial fibrosis [41,42]. It has also been observed that higher levels of TMAO in blood appear to be associated with a worse prognosis [43].

## 4. Diagnosis

The pathophysiological timeline of diabetic cardiomyopathy seems to follow the trend observed in other non-structural heart diseases, with the initial development of left ventricular diastolic dysfunction followed by subclinical systolic dysfunction with preserved ejection fraction and finally progressing to HFrEF [44,45]. In advanced stages, the diagnosis of HF is based on a combination of clinical data of the patient—compatible signs and symptoms based on the classic Framingham criteria—supported by diagnostic tests. Diagnostic confirmation is necessary in all cases, given its prognostic implication and the need to carry out an adequate therapeutic adjustment [46].

However, in the population with T1D, it is important to diagnose diastolic dysfunction and subclinical systolic dysfunction, to do an early diagnosis of the disease using sensitive cardiac markers that are easy to incorporate in routine clinical practice. Type B natriuretic peptides (BNP, NT-ProBNP) are plasmatic biomarkers, which are released in response to ventricular stretching and volume overload within the cardiac chambers, and can be affected by parameters such as age, sex, BMI, or renal function. These markers are a useful tool to guide the diagnosis of HF in the acute setting, in either diabetic or non-diabetic patients. Data from the multinational Breathing Not Properly trial suggest that diabetes is not a confounding variable in the interpretation of BNP levels in this situation [47]. A recent study [48] determined that higher NT-ProBNP levels were independently associated with HF in 664 subjects with T1D [HR 1.7 (95% CI: 1.1–2.4), *p* = 0.01]. The latest guidelines for the diagnosis and treatment of HF recommend the use of these natriuretic peptides both in acute and non-acute settings to rule out HF, given its high negative predictive value, but not to establish its diagnosis. Thus, the diagnosis in diabetic patients in the non-acute setting should follow the diagnostic algorithm that emphasizes that patients with a high probability of HF may have an echocardiogram to confirm or rule out the diagnosis [49]. Echocardiography is postulated as a central tool in the diagnosis of HF, given its safety, easy access, and highly informative character (cardiac chamber volumes, ventricular and valve function, and myocardial wall thickness, among other aspects) [50].

For the study of diastolic dysfunction in young people with T1D [51], it is recommended to follow the general indications of the American Society of Echocardiography and the European Association of Cardiovascular Imaging, through indices that include involving pulse Doppler transmitral inflow velocities (E and A waves) and tissue Doppler early and late mitral annular diastolic velocities (e′ and a′), atrial size measurements, and pulmonary venous flow evaluation [52]. Thus, in recent years, more sensitive ultrasound techniques have been incorporated to detect the more subtle abnormalities of cardiac function that would go unnoticed with conventional techniques and measurements (such as ventricular deformation and desynchrony indices). Although left ventricular diastolic dysfunction is the earliest manifestation of HF in the diabetic population [53], recently, the role of left atrial dysfunction as an active contributor to the initial diastolic dysfunction suffered by these patients has been revealed [54]. Ifuku M et al. [14] observed left atrial dysfunction (such as left Atrial phasic strain) in adolescents and young people with T1D (*n* = 53) compared to non-diabetic controls (*n* = 53) and assert that it could constitute an early and sensitive marker of diastolic dysfunction in T1D. The E/e′ ratio is frequently used as a marker of diastolic dysfunction (Yoldaş T, 2018). Bradley TJ et al. [55] observed an E/e′ ratio (7.3 ± 1.2 vs. 6.7 ± 1.3; *p* = 0.0003) increased in patients with T1D (*n* = 199) compared to non-diabetic subjects (*n* = 178). However, not all the findings are consistent in this regard [56].

Kaushik A et al. [57], in a recent study, found the presence of preclinical ventricular dysfunction echocardiographic alterations in the population with T1D. Specifically, they found lower left ventricular strain indices [basal lateral LV (21.39 ± 4.12 vs. 23.78 ± 2.02; *p* = 0.001), mid-lateral LV ( 21.43 ± 4.27 vs. 23.17 ± 1.92 *p* = 0.02), basal septum (20.59 ± 5.28 vs. 22.91 ± 2.00; *p* = 0.01), and mid septum (22.06 ± 4.75 vs. 24.10 ± 1.99; *p* = 0.01] in children and adolescents with T1D (*n* = 50) compared to non-diabetic controls (*n* = 25), despite the absence of manifest heart failure and normal ejection fraction. In addition, greater endothelial dysfunction was detected by flow-mediated dilatation (FMD) in subjects with T1D compared to non-diabetic patients (8.36 ± 4.27 vs. 10.57 ± 4.12, *p* = 0.04). These myocardial alteration parameters correlated with HbA1c levels (r = −0.327, *p* = 0.017). These findings reinforce the hypothesis of the possible early effect of the diabetic metabolic environment on myocardial function.

Some studies that evaluate systolic function in T1D with HFrEF expose a parallel reduction in ultrasound parameters such as longitudinal tension and global ventricular circumference, as well as a reduction in the systolic strain rate using speckle-tracking echocardiography [58], although not all studies have reported changes in this regard [59,60]. Different studies have also reported subclinical cardiac dysfunction in young subjects with T1D [61,62,63] although other studies have not reached this conclusion [60,61]. This controversy could probably be due to differences in the characteristics of the subjects with T1D -glycemic control, time of evolution of the disease [64], and the use of different ultrasound protocols for the determination of ultrasound parameters in the comprehensive evaluation of cardiac dysfunction, which calls for standardized approaches to facilitate their interpretation.

These basic and central examinations based on clinical, analytical, and mainly ultrasound parameters can be completed with other modalities such as cardiac magnetic resonance imaging. Cardiac MRI allows for calculating improved rates of myocardial deformation of diastolic incoordination, including biventricular desynchrony and incoordination. The EMERALD study, carried out in a young population with T1D, reports alterations in the diastolic pressure of the ventricular septum and the diastolic relaxation fraction, which reflects an uncoordinated and energetically less favorable myocardial relaxation compared to non-diabetic subjects [65].

Although the identification of this underlying heart problem in T1D can be very important to delay or prevent the development of manifest HF, it is necessary follow-up with these patients, through longitudinal studies, to accurately determine the clinical importance of the preclinical myocardial changes detected in this population.

## 5. Treatment

According to the latest European guidelines for the treatment and diagnosis of HF [49], in patients with HFrEF, interventions that reduce morbidity and mortality confer a similar benefit in the presence or absence of diabetes. In addition to the control of classic cardiovascular risk factors, the use of beta-blockers, angiotensin-converting enzyme inhibitors (ACEIs), spironolactone, or eplerenone is proposed. Meanwhile, other drugs are recommended only in selected patients with symptomatic HFrEF, both diabetics and non-diabetics, as is the case with the use of diuretics, sacubitril/valsartan, ivabradine, hydralazine, isosorbide dinitrate, or angiotensin II type I receptor blockers.

The identification of asymptomatic T1D patients with cardiac dysfunction may favor the development of useful therapeutic strategies in diabetic cardiomyopathy, to optimize the treatment of these patients and improve the prognosis of the disease. As we mentioned before, McAllister et al. [6] and Kristófi et al. [7] found that the total age-adjusted CVRD burden and risks were greater among patients with T1D compared with those with T2D and HF rates were significantly higher in T1D patients depending on the age group. They also highlighted that the use of antihypertensive, antiplatelet, and statin drugs was much higher in T2D than in T1D, although these differences could be explained by differences in age and comorbidities. These findings highlight the need to improve preventive strategies beyond glycemic control in the T1D population from an early age.

The gold-standard treatment in T1D is the use of basal-bolus insulin therapy and the early intensive therapy is a fundamental aspect to reduce HF risk in the long-term [13]. Currently, new strategies to measure glucose levels, including the detection of interstitial glucose through Continuous Glucose Monitoring (iCGM) or Flash Glucose Monitoring (FGM), allow the adjustment of insulin therapy to improve metabolic control and achieve optimal control, as well as a more accurate assessment of glycemic variability and its reduction [66,67]. Besides, because glycemic variability is an independent risk factor for developing long-term complications in diabetic patients, continuous glucose monitoring might be a valuable tool in this context [67].

Certain drugs approved for the treatment of T2D such as metformin, aGLP1, and iSGLT2 are being evaluated as potential complementary drugs to insulin therapy in T1D [68]. Reflections in this regard underline the importance of the proper selection of patients with T1D and a close follow-up of them, in the case of the use of iSGLT2 due to the associated risk of developing “normoglycemic” diabetic ketoacidosis (DKA) [69]. Based on recent positive results from the DEPICT study [70], dapagliflozin 5 mg was the first iSGLT2 to have its marketing authorization in Europe in March 2019 as an additional drug to insulin therapy in patients with T1D with a body mass index (BMI) ≥27 kg/m^2^ [71]. It also recently received Scottish Medicines Consortium (SMC) approval [72] as well as National Institute for Health and Care Excellence (NICE) approval following health economic analysis, in which dapagliflozin was found to be a highly cost-effective treatment option in people with T1D inadequately controlled by insulin alone [73]. However, its non-authorization in other places such as the U.S., and the BMI restrictions reflect safety concerns regarding the “normoglycemic” DKA risk. Since the available data are unclear, it is important to proceed on an individual basis for people that fall into these categories. Selecting appropriate people with T1D for iSGLT2 treatment is critical for minimizing the DKA risk and maximizing the potential benefits associated with this treatment. Those most likely to benefit from dapagliflozin treatment include overweight/obese people, established on stable optimized insulin therapy (i.e., not recently diagnosed), with high insulin needs (i.e., > 0.5 units/kg of body weight/day), and a low DKA risk profile, who have demonstrated adherence to their insulin regimen and the ability to understand and utilize relevant education relating to DKA risk [74].

In recent years, the development of these hypoglycemic molecules such as aGLP1 or iSGLT2 and the performance of cardiovascular safety studies for their commercialization have shown that they are not only beneficial in glycemic control, but also have cardioprotective effects in both T2D and non-diabetic patients. This opens the door to a clinical entity with important clinical repercussions, highly prevalent as we have seen in the general diabetic population and T1D. Furthermore, in certain stages, the lack of therapeutic options stands out, and although the studies show promising results, there are no specific data on the use of these drugs in the T1D population.

Metformin is the first-line treatment in T2D. In recent years, cohort studies and systematic reviews have analyzed its role in cardiovascular disease, finding that metformin seems to be associated with a reduction in mortality from all causes in T2D patients with HF, as well as with a reduction in readmissions by HF [75,76], so it is recommended in the current guidelines of the European Society of Cardiology [49] as a first-line drug in patients with T2D and HF. In T1D, REMOVAL a placebo-controlled trial to Metformin, data suggest that it might have a wider role in cardiovascular risk management, but do not support the use of metformin to improve glycemic control in adults with long-standing T1D [77].

Relative to iSGLT2, in the DECLARE-TIMI 58 trial, dapagliflozin treatment was associated with a lower rate of HF-related death and hospitalization than the placebo [78]. Likewise, dapagliflozin treatment has also been associated with a reduction in HF-related hospitalization rates in patients with or without HFrEF and a reduction in cardiovascular mortality and all-cause mortality compared to the placebo in patients with T2D and HFrEF [79], as well as in patients with T2D and previous myocardial infarction [80]. These benefits are the same in patients without diabetes with HFrEF [81], so its cardiovascular benefit would be independent of the hypoglycemic effect. The cardioprotective effects of empagliflozin are very similar [82]. These data are reinforced by later trials such as EMPRISE, where empagliflozin showed greater efficacy in the incidence of HF compared to sitagliptin, in reducing hospitalization for HF in T2D patients with and without cardiovascular disease [83]. Moreover, in the EMPEROR-Reduced trial, the use of empagliflozin reduced the risk of hospitalization for HF and cardiovascular mortality, regardless of the presence or absence of diabetes [84]. In this line, the EMPA-TROPISM (ATRU-4) study supports the benefit of empagliflozin in the treatment of HF regardless of its glycemic status, by demonstrating significant improvement in the key parameters of cardiac dysfunction, such as left ventricular (LV) volume, LV mass, LV systolic function, functional capacity, and quality of life of non-diabetic HFrEF patients [85]. Finally, results from the EMPEROR-preserve trial have been recently published, showing a reduction of the combined risk of cardiovascular death or hospitalization for HF with 10 mg empagliflozin in patients with HFpEF, regardless of the presence or absence of diabetes [86]. Other molecules of this pharmacological group have also shown benefits concerning cardiovascular mortality and hospitalization for HF (canagliflozin, CANVAS) [87], (sotagliflozin, SOLOIST-WHF) [88] or (ertugliflozin, VERTIS) [89]. In the case of sotagliflozin, they found benefits in HFpEF as well.

Regarding aGLP1, trials have shown heterogeneous information with favorable results in the reduction of cardiovascular mortality events for some molecules (Table 1): LEADER (liraglutide) [90], SUSTAIN-6 (semaglutide) [91], REWIND (dulaglutide) [92]; and neutral effects for other: ELIXA (lisixenatide) [93] and EXSCEL (long-acting exenatide) [94], without finding favorable specific results on HF. Subsequently, more specific trials have been conducted with liraglutide in patients with or without diabetes and HFrEF, which have further increased the uncertainty about the use of this molecule in subjects with established HF. The FIGHT trial, carried out in 300 patients recently hospitalized for HF, found that the use of liraglutide did not lead to greater clinical stability after hospitalization. Likewise, in the LIVE study (*n* = 241), it was found that the use of liraglutide did not affect left ventricular systolic function (LVEF) compared to the placebo in patients with stable HF, although it was associated with an increase in heart rate and serious adverse cardiac events, such as sustained ventricular tachycardia, atrial fibrillation, or worsening ischemic heart disease (10% vs 3%, *p* = 0.04). A meta-analysis published in recent years showed encouraging results regarding cardiovascular safety with the use of aGLP1, suggesting that they can reduce major adverse cardiovascular events, cardiovascular mortality, and all-cause mortality risk; no significant effect was identified in relation to hospitalization for HF [95] or even with reduced risk of hospitalization for HF [96]. A double-blind clinical trial [97] performed on T2D patients (*n* = 49) showed that treatment for 26 weeks with liraglutide versus a placebo reduced early diastolic LV filling and LV filling pressure to normal levels, pathogenic characteristics of HFpEF. However, future studies are needed to investigate these potential effects of aGLP1 in HF in its early stages and its benefits in other populations such as non-diabetic, obese, or T1D subjects.

Regarding the position of the different international cardiology and endocrinology societies in the use of iSGLT-2 and aGLP1, both are the recommended therapies in cases of T2D with cardiovascular disease, preferably leaning towards the use of the former in HF cases without ruling out the use of aGLP1 [68,98,99,100,101]. However, the American Heart Failure Society specifies the precaution of its use in situations of acute decompensation [102].

## 6. Conclusions

HF is a complication of increasing concern in diabetes, and given the high incidence of HF and the risk of hospitalization for HF in the population with T1D, more studies should be developed in this regard to clarify pathophysiological aspects, determine specific risk factors to control, and develop standardized protocols to establish specific precision biomarkers for the diagnosis of this entity in T1D patients from early stages.

The relationship between classic cardiovascular risk factors—such as hyperglycemia, hypertension, or dyslipidemia—and the cardiac and vascular abnormalities seen in people with T1D is not fully understood, so further research is required to identify potential treatment targets allowing for the development of therapeutic agents in this field. Some therapeutic groups, such as iSLGT2 and aGLP1, have shown a clear benefit in preventing cardiovascular complications in T2D. In particular, iSGLT2 have shown to be very effective in reducing HF-related deaths and hospitalization for HF in both T2D and non-diabetic patients. However, it remains to be determined if they are useful and safe in patients with HF and T1D.

## Figures and Tables

**Figure 1 jcm-10-04497-f001:**
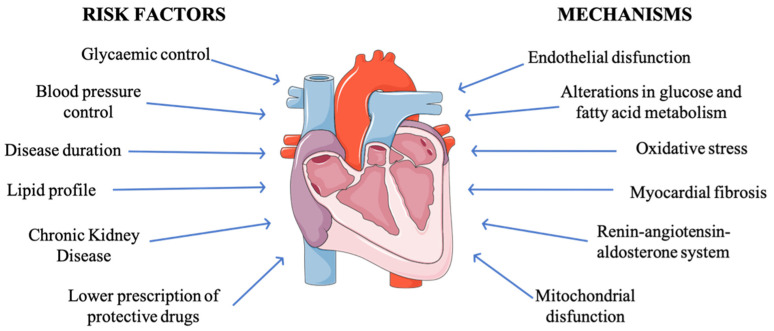
Risk factors and pathophysiology of heart failure in diabetes.

**Table 1 jcm-10-04497-t001:** Evidence on hospitalization for heart failure and cardiovascular mortality with glucagon-like peptides 1 agonist (aGLP1) and sodium-glucose cotransporter 2 inhibitor (i-SGLT2) from randomized controlled trials.

		Study	Hospitalization for HF	CV Mortality
**GLP1 receptor agonists**	Liraglutide	LEADER [90]		↓
	Semaglutide	SUSTAIN-6 [91]		↓
	Dulaglutide	REWIND [92]		↓
**SGLT2 inhibitors**	Dapagliflozin	DECLARE-TIMI 58 [78]	↓	
	Empagliflozin	EMPRISE [83]	↓	
		EMPEROR-Reduced [84]	↓	↓
		EMPEROR-Preserve [86]	↓	↓
	Canagliflozin	CANVAS [87]	↓	↓
	Ertugliflozin	VERTIS [89]	↓	

GLP1: glucagon-like peptide 1; SGLT-2: sodium-glucose cotransporter 2; HF: heart failure; CV: cardiovascular.

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
