# Peer review of "Heart Failure in Type 1 Diabetes: A Complication of Concern? A Narrative Review"

_jcm, 2021, doi:10.3390/jcm10194497_

Round 1

Reviewer 1 Report

Thank you for giving me a chance to review this submission to  your journal

This is a very timely article for a general internal medicine readership.  As the authors point out, heart failure in type 1 diabetes is increasing as our patients with type 1 diabetes live longer.  The etio-pathology in type 1 is less well understood then what it is believed to be in patients with type 2 diabetes. While some of these etio-pathologies are shared between the 2 conditions, there are some unique considerations for patients with type 1 diabetes and heart failure

In this paper, the epidemiology and especially the differences between types of diabetes and genders is concisely covered and appropriately documented and referenced. However, the DCCT/EDIC data is very important and needs to be fleshed out somewhat; while the authors accurately point out that every 1% increase in hemoglobin A1c is associated with a significant increase in the incidence of heart failure I think it is very important to point out that in DCCT early intensive control was associated with a significant decrease in the incidence of heart failure years later [Diabetes Care: 2016; 39, 686]

An emerging risk factor in type 1 diabetes, extensively reviewed in specific and specialized heart journals, but seemingly deemed not as yet ready for prime time among internal medicine readerships (don't know why!!) is the concept of cardiac autoimmunity induced by hyperglycemia [Circulation. 2020; 141: 1107-09] While this is only one reference there are a whole slew of articles on the subject and it is important to address this within the body of this article since it has implications for treatment in the early phases of the disease.  It would seem that the occurrence of this autoimmunity may be prevented by early tight glycemic control and/or maintaining low glycemic variability in the crucial early stages of the disease. Since the burden of the control of diabetes in its early stages is at least shared if not mainly dependent on internists, this early control needs to be stressed. The section on pathophysiology is otherwise well written and referenced  and informative

The diagnostic section is well written and introduces the utilization of BMP and NT pro BNP. This concept is again not generally recognized as a valid potential marker for heart failure in the general internal medicine readership and could prove crucial for early detection and treatment to stop progression

In the treatment portion once again the mention of glycemic variability and how this might be tested and established using CGM may be a worthwhile add on at least as a summary

The SGLT2 2 inhibitor section is very comprehensive but requires some careful thought and needs to address some possible concerns. At least in the USA, SGLT 2 inhibitors are not as yet licensed for use in type I diabetes. I understand that this is not the case in Europe. It would be opportune to explain the reason for this dichotomy viz  the concerns about the development of "normoglycemic DKA" with SGLT-2 use in type 1 diabetes. The pathophysiology can be summarized ( rise in Glucagon with SGLT2 inhibitors and the lowering of exogenous insulin doses to protect against hypoglycemia)  and strategies to prevent this complication can be briefly discussed since it can occur in both types of diabetes

The GLP-1 section is a very good summary

Overall I think this paper is appropriate for publication and could prove very informative to a general internal medicine readership who have not, as the authors point out, had adequate coverage of the problem of  heart failure in type 1 diabetes

I would like to point out that the suggestions proposed above are just suggestions and may reflect my prejudice in terms of explaining pathophysiology to make decisions about therapy logical and appropriate I would again thank you for this opportunity and I hope the above meets your requirements for your review

Author Response

Dear Editor and Reviewers,

We would like to thank you very much for your constructive comments and suggestions which have undoubtedly helped us to improve our manuscript.

In this new version of the manuscript, we have taken into consideration the comments and suggestions from reviewers and have revised the paper accordingly. We have responded to each of the reviewers' comments and have edited the manuscript to address all the reviewers' issues and suggestions.

We have provided the replies to the comments in the following section and have highlighted changes in the manuscript using the “Track Changes”.

We hope that our revised manuscript may now be found acceptable for publication in the journal. Nevertheless, we are of course willing to revise it further according to any other suggestions or concerns raised by the Editor or the Reviewers.

Yours faithfully,

Isabel Cornejo-Pareja, and coauthors

Reviewer Comments to Authors

Reviewer 1

Thank you for giving me a chance to review this submission to your journal.

This is a very timely article for a general internal medicine readership.  As the authors point out, heart failure in type 1 diabetes is increasing as our patients with type 1 diabetes live longer.  The etio-pathology in type 1 is less well understood then what it is believed to be in patients with type 2 diabetes. While some of these etio-pathologies are shared between the 2 conditions, there are some unique considerations for patients with type 1 diabetes and heart failure.

Comments:

In this paper, the epidemiology and especially the differences between types of diabetes and genders are concisely covered and appropriately documented and referenced. However, the DCCT/EDIC data is very important and needs to be fleshed out somewhat; while the authors accurately point out that every 1% increase in hemoglobin A1c is associated with a significant increase in the incidence of heart failure I think it is very important to point out that in DCCT early intensive control was associated with a significant decrease in the incidence of heart failure years later [Diabetes Care: 2016; 39, 686]

Response:

First, we would like to thank the reviewer for his/her thorough and constructive review of our manuscript and for offering us comments and suggestions to improve it. Regarding to DCCT data, we have included some additional information taken from the suggested reference to clarify the potential benefit of an early intensive care in the risk of heart failure in the long-term.

“Cardiovascular risk factors are well established and data from the Diabetes Control and Complications Trial (DCCT) and its observational follow-up Epidemiology of Dia-betes Interventions and Complications (EDIC) demonstrated that an early intensive pe-riod of 6–7 years of intensive glycemic control significantly reduced the risk for cardio-vascular disease. And though the effect tends to decrease over the years it remains highly significant 30 years later (a reduction of 30% compared to conventional treatment; p0.016)”.

“Therefore, the early intensive therapy seems to have an effect in reducing HF risk in the long-term (five-fold difference among intensive therapy group and conventional treatment), though in the analysis at 30 years the number of events related to HF were small to establish a definitive conclusion”.

Comments:

An emerging risk factor in type 1 diabetes, extensively reviewed in specific and specialized heart journals, but seemingly deemed not as yet ready for prime time among internal medicine readerships (don't know why!!) is the concept of cardiac autoimmunity induced by hyperglycemia [Circulation. 2020; 141: 1107-09] While this is only one reference there are a whole slew of articles on the subject and it is important to address this within the body of this article since it has implications for treatment in the early phases of the disease.  It would seem that the occurrence of this autoimmunity may be prevented by early tight glycemic control and/or maintaining low glycemic variability in the crucial early stages of the disease. Since the burden of the control of diabetes in its early stages is at least shared if not mainly dependent on internists, this early control needs to be stressed.

Response:

Thanks to the reviewer for the suggestion, we had not been aware of this line of investigation, and it has been very constructive and interesting to deepen in this field. Some comments to the reference suggested and some others found in our research have been included in the text, in the section of risk factors and pathophysiology.

Another emerging field of investigation in HF in T1D and that may be intimately linked to glycemic control is cardiac autoimmunity. In an analysis derived from DCCT/EDIC, Sousa et al measured the prevalence and profiles of cardiac autoantibodies in samples from DCCT and divided them in two groups, patients with HbA1c>9% (n=83) and patients with HbA1c<7% (n=83) at 26 years of follow-up. The same analysis was performed in similar groups of patients with T2D. They found that DCCT HbA1c >9% group had significant higher levels of cardiac autoantibodies than the DCCT HbA1<7% group, while glycemic control was not related to cardiac autoimmunity in T2D. Moreover, positivity for 2 or more autoantibodies during DCCT was associated with a greater risk of cardiovascular disease (HR 16.1 [95% CI 3.0–88.2]) and coronary artery calcification (OR 60.1 [95% CI 8.4–410.0]). Same author published recently that cardiac autoimmunity is related to subclinical myocardial dysfunction independently on classical cardiovascular disease risk factors. They observed in a sample from DCCT that patients with 2 or more cardiac autoantibodies had greater left ventricular end-diastolic volume, end-systolic volume, left ventricular mass, and lower left ventricular ejection fraction. These observations suggest that there are different mechanisms for cardiovascular disease and thus for HF in T1D and T2D and those mechanisms may be tightly related to long exposure to hyperglycemia in T1D.

Comments:

The section on pathophysiology is otherwise well written and referenced and informative.

 Response:

Thanks to the reviewer for the comment. We have included some insights for cardiac autoimmunity also in this section.

Regarding the immune system, as we mention before, Sousa et al observed higher levels of cardiac autoantibodies in patients with T1D and poor glycemic control, and those patients positive for ≥2 cardiac autoantibodies were more likely to have subclinical myocardial dysfunction as well as a higher cardiovascular disease risk. Chronic hyperglycemia may cause subclinical myocardial injury favoring the exposure of heart muscle proteins as a-myosin to the immune system. In patients with T1D and poor glycemic control the immune system is dysregulated and may overreact to these proteins, producing an expansion of proinflammatory CD4 T-cells specific to a-myosin and the development of autoantibodies.

Comments:

The diagnostic section is well written and introduces the utilization of BMP and NT pro BNP. This concept is again not generally recognized as a valid potential marker for heart failure in the general internal medicine readership and could prove crucial for early detection and treatment to stop progression.

Response:

Thanks to the reviewer for the comment.

Comments:

In the treatment portion once again the mention of glycemic variability and how this might be tested and established using CGM may be a worthwhile add on at least as a summary.

Response:

Thanks to the reviewer for the comment. We have included some insights for CGM also in this section.

The gold-standard treatment in T1D is the use of basal-bolus insulin therapy and the early intensive therapy is a fundamental aspect to reduce HF risk in the long-term [13]. Currently, new strategies to measure glucose levels, including the detection of intersti-tial glucose through Continuous Glucose Monitoring (iCGM) or Flash Glucose Moni-toring (FGM) allows the adjustment of insulin therapy to improve metabolic control and achieve optimal control, as well as more accurate assessment of glycemic variabil-ity and its reduction [66,67]. Besides, because glycemic variability is an independent risk factor for developing long-term complications in diabetic patients, continuous glucose monitoring might be a valuable tool in this context [67].

Comments:

The SGLT2 2 inhibitor section is very comprehensive but requires some careful thought and needs to address some possible concerns. At least in the USA, SGLT 2 inhibitors are not as yet licensed for use in type I diabetes. I understand that this is not the case in Europe. It would be opportune to explain the reason for this dichotomy viz the concerns about the development of "normoglycemic DKA" with SGLT-2 use in type 1 diabetes. The pathophysiology can be summarized (rise in Glucagon with SGLT2 inhibitors and the lowering of exogenous insulin doses to protect against hypoglycemia) and strategies to prevent this complication can be briefly discussed since it can occur in both types of diabetes. The GLP-1 section is a very good summary.

Response:

We would like to thank the reviewer for the suggestions about this section, since they have allowed us to improve it by focusing certain controversial aspects. We have included additional information extracted that speaks of the points to take into account for the selection of patients in the use of iSGLT-2 in DM1 to potential the positive effects and minimize complications.

Based on recent positive results from the DEPICT study [70], dapagliflozin 5 mg has been the first iSGLT2 to have its marketing authorization in Europe in March 2019 as an additional drug to insulin therapy in patients with T1D with a body mass index (BMI) ≥27 kg/m2 [71]. It has also recently received Scottish Medicines Consortium (SMC) approval [72], and additionally has National Institute for Health and Care Excellence (NICE) approval following health economic analysis, in which dapagliflozin was found to be a highly cost-effective treatment option in people with T1D inadequately controlled by insulin alone [73]. However, the non-authorization yet in other places such as the USA, and the BMI restrictions reflects safety concerns regarding the "normoglycemic" DKA risk. Since the available data are unclear, it is important proceeding on an individual basis for people that fall into these categories. Selecting appropriate those people with T1D for iSGLT2 treatment is critical for minimizing the DKA risk and maximizing the potential benefits associated with this treatment. Those most likely to benefit from dapagliflozin treatment include overweight/obese people, established on stable optimised insulin therapy (i.e. not recently diagnosed), with high insulin needs (i.e. > 0.5 units/kg of body weight/day), and a low DKA risk profile, that have demonstrated adherence to their insulin regimen and the ability to understand and utilise relevant education relating to DKA risk [74].

Final comments

Once again, we are grateful for the opportunity to revise and improve our manuscript.

We hope that our revised manuscript may now be considered acceptable for publication in the journal. Nevertheless, we are of course willing to revise it further according to any other suggestions or concerns raised by the Editor or the Reviewers.

Yours faithfully,

Isabel Cornejo-Pareja, and coauthors

Reviewer 2 Report

Overall well written review.

  • Please review for several grammatical errors.
  • Are all the mechanisms in the pathophysiology section related to T1DM and HF or T2DM and HF or both? Please specify which mechanisms are related to each.
  • Please expand more on the EMPEROR Preserved trial, especially given recent findings.

Author Response

Dear Editor and Reviewers,

We would like to thank you very much for your constructive comments and suggestions which have undoubtedly helped us to improve our manuscript.

In this new version of the manuscript, we have taken into consideration the comments and suggestions from reviewers and have revised the paper accordingly. We have responded to each of the reviewers' comments and have edited the manuscript to address all the reviewers' issues and suggestions.

We have provided the replies to the comments in the following section and have highlighted changes in the manuscript using the “Track Changes”.

We hope that our revised manuscript may now be found acceptable for publication in the journal. Nevertheless, we are of course willing to revise it further according to any other suggestions or concerns raised by the Editor or the Reviewers.

Yours faithfully,

Isabel Cornejo-Pareja, and coauthors

Reviewer Comments to Authors

Reviewer 2.

Comments:

Overall well written review.

Please review for several grammatical errors.

Response:

First, thanks to the reviewer for the constructive comments and suggestions. We have reviewed the full manuscript and corrected English spelling and grammar. Changes are highlighted by the “Track Changes” option.

Comments:

Are all the mechanisms in the pathophysiology section related to T1DM and HF or T2DM and HF or both? Please specify which mechanisms are related to each.

Response:

Thanks to the reviewer for the recommendation. Although HF is more investigated in T2DM, in this review we focus in pathophysiology mechanisms studied in T1DM. Referenced articles have been selected from those including T1DM patients or insulinopenic diabetes mouse models. Nevertheless, several mechanisms are common for both types. We have added some clarification in order to specify which of those mechanisms are shared.

Diabetic cardiomyopathy pathophysiology is widely studied in T2D, while mechanisms in T1D are less clear. Hyperglycemia and chronic inflammation present in both types promote cardiac hypertrophy and fibrosis, rising myocardial stiffness and resulting in diastolic and systolic dysfunction.

Comments:

Please expand more on the EMPEROR Preserved trial, especially given recent findings.

Response:

In this section, we have included additional information extracted to EMPEROR Preserved trial recently published (27/08/2021).

Finally, results from the EMPEROR-preserve trial have been recently published, announcing a reduction of the combined risk of cardiovascular death or hospitalization for HF with 10mg empagliflozin in patients with HFpEF, regardless of the presence or absence of diabetes [86].

Final comments

Once again, we are grateful for the opportunity to revise and improve our manuscript.

We hope that our revised manuscript may now be considered acceptable for publication in the journal. Nevertheless, we are of course willing to revise it further according to any other suggestions or concerns raised by the Editor or the Reviewers.

Yours faithfully,

Isabel Cornejo-Pareja, and coauthors

Round 2

Reviewer 2 Report

The authors have improved on the corrections initially sent for revision.